# Epithelial Mesenchymal Transition (EMT) and Associated Invasive Adhesions in Solid and Haematological Tumours

**DOI:** 10.3390/cells11040649

**Published:** 2022-02-13

**Authors:** David Greaves, Yolanda Calle

**Affiliations:** School of Life Sciences and Health, University of Roehampton, London SW15 4JD, UK; greavesd@roehampton.ac.uk

**Keywords:** EMT, haematological tumours, cell adhesions, drug resistance, migration, invasion

## Abstract

In solid tumours, cancer cells that undergo epithelial mesenchymal transition (EMT) express characteristic gene expression signatures that promote invasive migration as well as the development of stemness, immunosuppression and drug/radiotherapy resistance, contributing to the formation of currently untreatable metastatic tumours. The cancer traits associated with EMT can be controlled by the signalling nodes at characteristic adhesion sites (focal contacts, invadopodia and microtentacles) where the regulation of cell migration, cell cycle progression and pro-survival signalling converge. In haematological tumours, ample evidence accumulated during the last decade indicates that the development of an EMT-like phenotype is indicative of poor disease prognosis. However, this EMT phenotype has not been directly linked to the assembly of specific forms of adhesions. In the current review we discuss the role of EMT in haematological malignancies and examine its possible link with the progression towards more invasive and aggressive forms of these tumours. We also review the known types of adhesions formed by haematological malignancies and speculate on their possible connection with the EMT phenotype. We postulate that understanding the architecture and regulation of EMT-related adhesions will lead to the discovery of new therapeutic interventions to overcome disease progression and resistance to therapies.

## 1. Introduction

Epithelium integrity is maintained by the apical-basal polarity of epithelial cells generated by adhesions at cell-cell junctions and with the basal lamina. However, under specific physiological conditions, epithelial cells lose the contacts with neighbouring cells and the subjacent matrix, adopting a highly motile mesenchymal phenotype. This cell behaviour is called epithelial-mesenchymal transition (EMT) and it is critical for tissue morphogenesis during embryonic development and in adulthood for wound healing [1,2]. At the molecular level, EMT is characterised by the downregulation of the cell adhesion protein E-cadherin, leading to the disassembly of intercellular junctions and the initial dissociation of epithelia. In parallel, cells undergo major cytoskeletal rearrangements involving the downregulation of keratins and the upregulation of vimentin, which facilitates the highly migratory behaviour in the dissociated cells [2]. Several well characterised transcription factors (TFs) are upregulated during EMT such as SNAIL/SNAI1, SLUG/SNAI2, TWIST and the ZEB family of TFs, which orchestrate the downregulation of E-cadherin [3] and the upregulation and activation of N-cadherin and the intermediate filament vimentin [4]. Reciprocally, upregulated vimentin can promote the expression or the activity of EMT TFs and other signalling pathways favouring an invasive phenotype [5,6]. 

EMT is also thought to play a central role in cancer progression by facilitating the invasive behaviour of tumour cells, enabling tissue infiltration and the formation of metastatic foci [7,8]. Consequently, expression of EMT markers is associated with an aggressive malignant phenotype and correlates with the poor prognosis of cancer patients [8,9,10]. Over the last decade, development of an EMT phenotype in tumour cells in cancers of epithelial origin (carcinomas) has also been associated with acquisition of further malignant traits including cancer stem cell properties [11,12], immunosuppression [13,14], as well as resistance to therapeutic agents [15,16,17,18,19] and to radiotherapy [20,21]. 

The induction and maintenance of EMT is highly regulated by extracellular signals during embryonic development, tissue repair and tumour progression. These include chemical cues such as cytokines, chemokines, the presence of reactive oxygen species (ROS) [22] and hypoxia [23,24], as well as mechanochemical signals provided by changes in the rigidity of the surrounding extracellular matrix [25,26]. Specifically in tumours, autocrine and paracrine stimulation of cancer cells by cytokines such as TGFβ1 induce EMT while promoting an immunosuppressive microenvironment [27,28]. Additionally, macrophages and cancer associated fibroblasts present in the tumour microenvironment can secrete classical EMT-inducing cytokines/chemokines including HGF, SDF1α, EGF, PDGF as well as TGFβ1 [24,28]. Cytokines also secreted by the tumour stroma and firstly identified as pro-survival, immunosuppressive or as chemotactic signals to recruit immune cells to the tumour microenvironment are now known to also induce EMT in cancer cells. Some examples are IL-6, IL-8, IL-10 and TNFα [27,29]. The hypoxic microenvironment generated in growing tumours prior to vascularisation is another extensively studied factor that promotes EMT in cancer cells [23,30,31]. Hypoxia drives EMT by triggering various signalling pathways [32], the most commonly known being the stabilisation of Hypoxia Inducible Factor-1 α (HIF-1α) [33], that results in increased activity of the EMT TFs SLUG [33], SNAIL [32] and TWIST [34]. 

EMT can be reversed by a process known as mesenchymal-epithelial transition (MET). The coordination of EMT and MET is critical in both physiological and pathological conditions [2]. During the formation of metastatic foci, EMT facilitates the dissemination of cancer cells from the primary tumour to the blood and lymph circulation systems. After extravasation of circulating tumour cells to a secondary organ, they undergo MET to convert from a non- or low proliferative, migratory phenotype to a proliferative type that will generate a secondary tumour. However, the current scientific consensus in the field establishes that only under some very specific circumstances during development do epithelial cells switch completely from an epithelial to a fully mesenchymal phenotype [2]. It is currently agreed that there is a spectrum of possible intermediate stages characteristic of specific EMT programmes that encompass the loss of characteristic features of epithelial cells while acquiring the traits of mesenchymal cells, resulting in a more invasive/motile phenotype [2]. It remains undetermined whether the molecular and cellular EMT features in these programs comprise discrete stages of a linear spectrum of EMT- MET conversion or whether, instead, they represent a continuum of possible non-epithelial states expressed depending on the biological context [2]. 

In cancer cells, gene mutations and/or the influence of the tumour stroma may induce patterns of expression of epithelial and mesenchymal markers resulting in particular EMT phenotypes [2,10,35]. This epithelial/mesenchymal plasticity spectrum provides tumours with a landscape of possible phenotypes with multiple associated functions [36] and capacity for adaptation to various microenvironments [37,38]. For example, circulating tumour cells express characteristic intermediate epithelial-mesenchymal traits [37,38] whereas once the cells colonise a secondary site, they undergo a further MET conversion [39,40]. Additionally, certain EMT profiles have been associated with the subsequent expression of gene signatures of tumour subpopulations in skin and breast carcinomas that may allow these cells to colonise specific organs [36]. This possible control of gene expression by distinct EMT profiles may infer cancer-stem cell [41,42] and/or drug resistant traits to subpopulations of tumour cells [13,15,16,17,18,19], leading to refractory disease and relapse in cancer patients.

A substantial body of evidence supports the central role of EMT in the invasive behaviour and the metastatic disease of carcinomas such as lung, prostate, colon, liver, pancreatic and breast cancers [24]. However, in the last decade it has become evident that an EMT behaviour is also present in non-epithelial solid tumours of mesenchymal origin and in haematological malignancies [9,10,23,43]. This EMT-like behaviour in mesenchymal tumours may echo the loss of polarity and increased migratory capacity of endothelial cells during tissue repair and angiogenesis in an EMT-related process named endothelial-mesenchymal transition [1,44]. 

This review will emphasize the development of EMT-phenotypes during tumour progression of haematological malignancies. We will discuss the current understanding of the specific molecular EMT features described in haematological cancers and propose possible associated adhesomes formed by haematological tumour cells that may facilitate this invasive and aggressive phenotype.

## 2. EMT in Haematological Malignancies

Haematopoietic cells derive from the mesoderm and, therefore, have a mesenchymal developmental origin, however, the expression of EMT-like signatures with upregulation of canonical mesenchymal markers has been described in all types of haematological malignancies (lymphomas, multiple myeloma and lymphoid and myeloid leukaemia) [10]. The current literature strongly indicates a clear correlation between the expression of these EMT signatures, or genetic abnormalities in EMT TFs, and poor prognosis of patients [9,45,46,47,48,49,50,51]. One of the most common characteristics is the expression of high levels of vimentin in various forms of aggressive haematological tumours [9,52,53]. However, the exact biological role of the EMT markers in haematopoietic cancers remains largely unstudied. 

In particular, the actual correlation between the higher levels of EMT markers with a more motile phenotype, which is a key EMT feature, has not yet been studied in great depth in haematological malignancies in comparison to solid tumours. The majority of the studies have concentrated on various other tumour traits that may explain the correlation between expression of EMT signatures and the poor prognosis of patients. For example, the expression of EMT TFs has been associated with the repression of differentiation markers, correlating with more aggressive forms of haematological malignancies. Increased expression of ZEB1 promotes the methylation and downregulation of B-Cell Lymphoma protein 6 (BCL6), a key transcription factor that promotes differentiation of B cells and whose expression is associated with a benign profile [45]. A correlation has also been established between EMT and resistance to therapies. ZEB1 has been shown to promote drug resistance in mantle cell lymphoma by activating proliferation-associated genes while repressing pro-apoptotic ones and regulating the expression levels of membrane transporters involved in drug influx and efflux [46]. The upregulation of TWIST1 levels in chronic myeloid leukaemia (CML) promotes resistance to the therapeutic drug imatinib [50]. However, the cellular and molecular TWIST1-dependent mechanisms that may explain the lack of response to this drug have not yet been elucidated. Similarly, expression of EMT TFs in multiple myeloma cells in response to cues from the tumour microenvironment correlates with resistance to the clinical drugs dexamethasone and Velcade (bortezomib) [54]. 

Only a few of these studies have addressed the connection between the expression of EMT markers and increased cell migration. In paediatric anaplastic large cell lymphoma cells, expression of the oncogenic protein anaplastic lymphoma kinase (ALK) correlated with expression of higher levels of TWIST1. Downregulation of TWIST1 in these cells inhibited invasive migration and reinstated the efficacy of therapeutic drugs [49]. In MLL-AF9 driven acute myeloid leukaemia (AML), poor prognosis in patients correlated with expression of EMT markers and experimental downregulation of ZEB1 in AML cells inhibited the invasive capacity of this aggressive cancer [9].

Multiple myeloma is the haematological tumour where the EMT phenotype has been studied in more depth, and there is substantial evidence of the acquisition of a migratory phenotype orchestrated by the EMT program [23,43,51,54]. Multiple myeloma results from the accumulation of malignant plasma cells in disseminated tumour foci within the bone marrow. In 2012, the Ghobrial lab demonstrated that during tumour progression, hypoxia induced an EMT program in multiple myeloma cells that resulted in increased mobilisation and formation of further bone marrow foci [23]. This program was the result of the upregulation of HIF-1α leading to expression of SNAIL and downregulation of E-cadherin [23]. It was then proposed that multiple myeloma may be envisaged as a metastatic disease where progression results from continuing trafficking of myeloma cells from initial proliferating tumours in the bone marrow that acquire a hypoxic-driven EMT phenotype as they expand, leading to mobilisation of myeloma cells undergoing EMT [43]. Mobilised myeloma cells into the circulation express cell adhesion molecules such as β7 integrins [55] that would facilitate the interaction of circulating cells with the bone marrow endothelium and the extravasation into a new site. The re-entry into the bone marrow in this vascularised normoxic environment would repress the expression of HIF-1 α, silencing the EMT program and promoting a transition similar to MET in epithelial tumours, leading to the proliferation of myeloma cells and development of a new tumour [23]. Additional factors that may lead to the mobilisation of myeloma cells through the acquisition of an EMT-phenotype include the increased concentration of cytokines such as IL-6, TNFα, VEGF, HGF and IGF-1 in the tumour microenvironment during the symptomatic phase of the disease [43]. More recently, further evidence shows that cytogenetic abnormalities associated with a poor prognosis in multiple myeloma [51] as well as the composition of the extracellular matrix of the myeloma bone marrow microenvironment [54] promote an EMT invasive phenotype in these cancer cells.

Taken together, the studies in haematological malignancies indicate a compelling connection between expression of EMT markers and poor prognosis (Table 1). In addition, the acquisition of an invasive migratory phenotype has been corroborated in the limited number of in depth studies so far available (Table 1). However, the exact coordination between the levels of expression of EMT TFs and markers, and the migratory features of haematological cancer cells is practically unknown with exception of a few studies in multiple myeloma. More significantly, the exact adhesions and cytoskeletal configurations that may be assembled by haematological cancer cells undergoing EMT, have not been yet elucidated. In the next section, we review the structure, regulation and known roles of EMT-related adhesions in solid tumours and discuss the invasive adhesions so far described in haematological malignancies and their possible, although not yet proved, association with EMT.

## 3. Invasive Adhesions Formed by Cells Undergoing EMT in Solid Tumours: Are They Related to Adhesions Described in Haematological Malignancies?

Cell adhesion molecules and associated proteins, in addition to anchoring cells to the surrounding microenvironment and facilitating migration, are known to regulate gene expression and cell cycle progression [56,57,58,59]. For instance, integrin mediated signalling at adhesions modulates the activity of cell cycle check points such as the downregulation of CDK inhibitors and the induction of cyclin D during progression of the G1 phase [56,57,58,59]. Initiation of DNA synthesis requires sustained adhesion signalling [58,60] and transition from G2 to mitosis is also dependent on the level of formation of cell adhesions [61,62]. The recent finding of the direct binding between the cell cycle regulator CDK1 and talin, a key component of integrin-mediated adhesions, further reinforces a critical link between the organisation of cell adhesions and cell cycle progression [63]. Thus, during EMT the rearrangement of cell adhesions is likely to influence fundamental cell functions that determine the proliferative and invasive capacity of cancer cells and, therefore, tumour progression. Consequently, therapeutically targeting the signalling complexes at adhesion sites associated with EMT may not only repress the invasive capacity of cancer cells but it may also block survival and cell cycle progression, preventing tumour development and/or resistance to drug treatments [11,12,13,14,15,16,17,18,19,20,21]. 

We propose that to identify effective therapeutic targets for aggressive tumours, it is imperative to investigate the signalosomes at cell adhesions formed in cancer cells that have developed an invasive EMT phenotype. It is then essential to understand the architecture and regulation of EMT-related adhesions. 

### 3.1. Adhesions Formed in Solid Tumours Undergoing EMT

The onset of the EMT process requires the reorganisation of filamentous actin (F-actin) microfilaments and microtubules in coordination with the reticular assembly of intermediate filaments of upregulated vimentin, while cells acquire migration polarity. Polarised cells display a distinctive protrusive leading edge and a contractile rear end that facilitates motility. Polymerisation of F-actin is the driving force that pushes the cell membrane at the leading edge, forming finger like protrusions containing parallel bundles of linear actin filaments called filopodia, or sheet like extensions shaped by the formation of a branched network of F-actin, called lamellipodia. Filopodia are generally thought to detect and integrate chemotactic signals that will determine the directionality of the cell movement [64] and they can mechanically facilitate cell displacement [65,66]. Lamellipodia are responsible for the substantial protrusive activity at the leading edge of mesenchymal migrating cells. Interestingly, filopodia can transform into lamellipodia by initiating dendritic actin nucleation on originally linear actin filaments, showing that these structures cooperate and are inter-convertible [67].

Nascent actin filaments polymerising at the cell membrane are shaped by various families of actin binding proteins. Formins are responsible for the linear polymerisation of F-actin in filopodia while fascin cross-links the formed filaments in parallel bundles [68,69]. Formation of filopodia [70,71] downstream of activation of TWIST1 and SNAIL1 [72] and the expression of formins [73,74] regulate the migration of cancer cells undergoing EMT. Additionally, the upregulation of formins contributes to metastasis formation [75], and high levels of fascin correlate with poor prognosis in a number of cancers including breast, lung, gastrointestinal and oral squamous cell carcinoma [64,76,77], suggesting that their role in EMT contributes to cell invasion and the formation of secondary tumour foci. Increased expression of fascin is also associated with the development of EMT-mediated drug resistance in hepatocellular carcinoma [78].

Equally relevant for the development of an effective EMT phenotype in cancer cells is the formation of lamellipodia. F-actin assembly at lamellipodia is controlled by the Arp 2/3 complex, a multimeric protein that promotes lateral branching of polymerising actin filaments. Activation of the actin nucleation-promoting factors N-WASP/WASP and Scar/WAVE, downstream of the Rho-GTPases Cdc42 and Rac, respectively [79,80], regulates the activity of the Arp2/3 complex. The formation of lamellipodia in cancer cells undergoing EMT [81], together with the expression of high levels of the Arp2/3 complex and WAVE2 correlates with poor prognosis in several types of tumours [82,83] and strongly suggests a significant role of lamellipodia in the invasive migration of metastatic cells.

Both filopodia and lamellipodia need to be stabilised by clusters of cell adhesion molecules called focal contacts that dynamically tether the membrane on the surface of neighbouring cells or on the surrounding extracellular matrix. In cells undergoing EMT, the cell adhesion molecules identified at focal contacts include proteoglycans (characteristically CD44) and integrins [84]. These initial cytoskeletal and/or adhesion rearrangements may further develop into additional adhesive and protrusive structures such as invadopodia or microtentacles.

#### 3.1.1. Invasive Focal Contacts

Focal contacts that sustain filopodia and lamellipodia during EMT also show the capacity for local degradation of the extracellular matrix. CD44 clustered at focal contacts can form a complex with the metalloprotease (MMP) MT1-MMP at lamellipodia [85] suggesting a possible role of this receptor in organising and directing the dynamics of degradation sites. Additionally, CD44 creates a significant signalling node involved in EMT by regulating the activity of Src kinase [86]. Integrins have also been implicated in regulating the degradative function of focal contacts through the intracellular interaction with ILK, a kinase that facilitates the localised secretion of MMP9 [87].

Invasive focal contacts are also critical for the development of other cancer traits such as drug resistance. Targeting the cholesterol-rich domains (also called lipid rafts) that contribute to integrin clustering inhibits the signalling at adhesion sites, blocking migration while re-sensitizing drug-resistant cancer cells to therapeutic drugs [88]. These data reinforce the idea that targeting adhesions formed during EMT not only will prevent cancer cell migration but will also block the activation of signalling pathways involved in cancer cell survival, stemness and drug resistance.

#### 3.1.2. Invadopodia

Invadopodia are F-actin dependent adhesions that protrude at the ventral surface of a cell, belonging to a larger family of adhesions called invadosomes, which also include podosomes [89,90,91]. Whilst similar, podosomes and invadopodia are structurally distinct, and form in different cell types. Podosomes are typically found in cells of monocytic lineage, endothelial cells [91,92,93], smooth muscle cells [94], and neuronal growth cones [95]; invadopodia are only observed in tumour cells and Src-transformed fibroblasts [91,96]. Invadopodia attach cells to the extracellular matrix while focally secreting MMPs. This process facilitates penetration of the extracellular matrix leading to subsequent cell migration and tissue invasion [97,98,99]. Formation of invadopodia is closely associated with the development of an EMT phenotype, and their function is to facilitate the infiltration of cancer cells across tissue barriers such as the endothelial basal lamina in blood vessels [100,101]. The structure of invadopodia comprises a protrusive F-actin-rich core assembled by cortactin, N-WASP, and the Arp2/3 complex, which is linked to the plasma membrane by cell adhesion proteins including integrins and CD44 [96,97]. Unlike podosomes, targeted microtubules and vimentin polymerisation can drive further extension of mature invadopodia into the extracellular matrix [102].

There is recent evidence indicating that the formation of invadopodia is coupled to cell cycle progression and EMT. Expression of invadopodia constituents is increased in G1, correlating with the enhanced capacity for degradation of the extracellular matrix of cancer cells [103]. Additionally, induction of the cell cycle checkpoint kinase 2 (Chk2) leads to cell cycle arrest in G2/M and suppression of TWIST1, SNAIL1, ZEB1 and vimentin expression, resulting in inhibition of invadopodia formation and cancer cell invasion in p53-mutated cells [104]. It is tempting to speculate that, reciprocally, invadopodia formation may have an impact in the regulation of the cell cycle during EMT, since vimentin is a key constituent of invadopodia and it has been shown to regulate cell cycle progression through its interaction with 14-3-3 proteins [105,106]. Vimentin availability and its post-translational modifications at invadopodia may affect its capacity for sequestration of free 14-3-3 and the formation of complexes mediated by these proteins, which in turn would affect the transition in G1/S and G2/M phases of the cell cycle [107].

#### 3.1.3. Microtentacles

Cancer cells form microtubule-based cell long tubular membrane protrusions used for cell migration that receive myriad names within the established literature [108,109]. As these protrusions are all structurally and functionally related, in this review they will all be referred to as microtentacles (McTNs). McTNs are CD44 dependent membrane protrusions containing actin and vimentin in addition to microtubules [110,111]. McTNs are typically around 1 µm in diameter and allow for invasive migration within tissues [111]. They can also be assembled by detached circulating tumour cells [109], and they facilitate integrin-dependent adherence to the endothelium to penetrate endothelial cell-cell junctions [112,113]. Expression of MT1-MMP at McTNs facilitates degradation of endothelial cell-cell connections and the basement membrane allowing extravasation of the circulating tumour cells [114,115]. Although in some instances McTNs may be morphologically quite similar to canonical filopodia, they clearly differ structurally, as they contain vimentin, microtubules, and their actin content is cortical rather than central [116]. Additionally, actin polymerisation drives filopodia protrusion whereas the formation of McTNs requires a fine balance between microtubule and actin polymerisation, which promote McTNs extension and retraction, respectively [111]. Formation of McTNs has been linked directly to the development of EMT. Upregulation of TWIST1 downregulates the expression of the tubulin tyrosine ligase enzyme, resulting in an accumulation of detyrosinated α-tubulin. This leads to microtubule polymerisation, the extension of McTNs, and reattachment and infiltration of cancer cells across endothelial layers [113,117].

Assembly of McTNs and invadopodia are interchangeable in breast cancer cells. The activation of Src kinase promotes the diversion towards the formation of invadopodia, whereas the repression of Src activity leads to the assembly of McTNs [118]. This interconversion provides breast cancer cells with a plasticity to form metastatic foci that may result in drug resistance to Src inhibitors.

### 3.2. Adhesions Formed in Haematological Cancer Cells

In contrast to solid tumours, in haematological malignancies there is a paucity of published studies that clearly connect the development of the EMT phenotype with the formation of specific types of adhesions. However, there is strong evidence that haematological tumours can form adhesions that are structurally and/or functionally related to those formed in solid tumours undergoing EMT. The following adhesions below have been observed in haematological tumour cells and display close similarities to the EMT-related adhesions formed in solid tumours. We propose that these adhesions should be investigated in the future in the context of EMT-like processes in haematological malignancies.

#### 3.2.1. Invasive Focal Contacts

In general, haematological cancer cells in in vitro culture do not readily attach to plastic, and only a small proportion of cells form small and labile focal contacts on extracellular matrix proteins such as fibronectin by clustering integrins and associated proteins. These adhesions contain many components present in focal contacts assembled by cancer cells undergoing EMT in solid tumours including Src kinases, FAK, p130Cas, Cbl, Abi1 and CrkL [119,120,121], but they lack some of the integrin-binding proteins that confer adhesion stability such as paxillin and vinculin [119,120]. However, in response to activation with 12-O-Tetradecanoylphorbol-13-acetate (TPA), phorbol-12-myristate-13-acetate (PMA) or certain cytokines, paxillin and vinculin are incorporated into these initial focal contacts (Figure 1), correlating with the increased attachment of cell cultures [120]. These mature focal contacts have also been shown to be sites of secretion of metalloproteases such as MMP2 and MMP9 [121], suggesting a role for remodelling and/or degradation of the extracellular matrix. Focal contacts have been proposed to be involved in cell adhesion mediated drug resistance to clinical drugs by mediating the interaction of leukaemia and multiple myeloma cells with the surrounding tissue stroma [55,122]. They have also been shown to modulate cell division by facilitating the ILK-dependent orientation of the centrosomes and the organisation of the mitotic spindle [123], indicating a role in cell proliferation. Targeting the disassembly of focal contacts in combination with cytotoxic drugs could improve therapies for relapse and refractory leukaemia patients [124,125,126].

#### 3.2.2. Podosomes

Similarly to invadopodia, podosomes are sites of attachment and localised secretion of metalloproteinases that under normal physiological conditions mediate the migration of normal myeloid cells across tissue boundaries. In osteoclasts, podosomes facilitate migration on the bone surface and they can mature into an actin structure called the sealing zone that creates a confined area for secretion of acidic enzymes and proteases to dissolve the bone matrix. In vitro, TPA/PMA activated myeloid leukaemia cells can also form podosomes [127], which suggests a possible role of these structures in facilitating leukaemia cell motility. The factors that may determine the choice of formation of podosomes or focal contacts in similarly activated myeloid leukaemia cells remain to be investigated, as does whether these cell adhesions are interchangeable. Podosomes have been shown to facilitate the invasive motility of B-CLL cells across endothelial barriers [128]. In contrast to normal B-cells, which are completely devoid of podosomes, B-cell leukaemic cells can form degradative podosomes without any additional in vitro activation [128,129]. We have also observed the formation of podosomes with typical F-actin cores and a ring of integrin associated proteins such as talin in primary multiple myeloma cells seeded on fibronectin (Figure 2). Other authors have shown that myeloma cell lines and primary cells from myeloma patients can also be differentiated into osteoclastic cells and degrade the bone matrix, a feature that is not associated with normal B-plasma cells and that requires podosome formation [130,131]. 

The modulation of the signalosome in podosomes may not only affect the migratory capacity of haematological tumour cells but also the expression of genes involved in cell survival, proliferation and secretion of cytokines, critical mechanisms involved in cancer progression, immunosuppression and drug resistance. Several of the proteins involved in organising the F-actin architecture of podosomes, such as WASP [132] and the Arp2/3 complex [133,134], have also been shown to translocate to the nucleus where they regulate transcription of various cytokines [132] and the activity of RNA polymerase II [133], respectively. Interestingly, the regulation of podosomal components may be particular to haematological tumours in comparison to normal myeloid cells. For example, the phosphorylated form of the transcription factor STAT5 accumulates in podosomes formed by activated CML cells, whereas it localises to the nucleus in normal myeloid cells [135]. The exact role of the recruitment of STAT5 to podosomes is presently unknown, but it may regulate the expression of genes controlled by this TF. Taken together, the current studies suggest that in haematological malignancies, podosomes may facilitate invasive migration and they may be involved in the regulation of gene expression.

#### 3.2.3. Podia

Podia are tubular membrane projections formed by protrusive activity and sustained by integrins and CD44 mediated adhesions that are observed in haematopoietic stem cells [136,137] as well as in leukaemia [137] and multiple myeloma cells [138] (Figure 3). These structures may be reminiscent of the McTNs assembled by cells undergoing EMT in solid tumours. They may also be related to protrusive structures named magnupodia recently described in polarised migrating haematopoietic stem cells [136]. 

Chemotactic and EMT promoting signals such as SDF1-α induce podia formation in leukaemia [140] and haematopoietic stem cells [136]. Podia assembly requires the activation of the degradative enzyme elastase in leukaemia cells [140] but the possible extracellular matrix degradation function of podia has not been demonstrated. It is thought that podia may contribute to the egress and exit of leukaemia cells from the circulation, leading to the formation of distant foci [140]. Interestingly, inhibition of elastase activity in leukaemia cells inhibited cell migration and promoted proliferation [140], a phenotype similar to that of cells undergoing MET. However, no clear link between podia formation and any stages of the EMT process has been demonstrated. The detailed molecular structure, exact function and the regulation of podia in haematological malignancies is largely undefined. 

In summary, the possible connection between the formation of podia, invasive focal contacts or podosomes and the EMT phenotype in haematological malignancies remains to be elucidated. This is an emerging field of research in haematological tumours with significant potential to identify critical signalling nodes involved in invasive migration, metastatic behaviour, and drug resistance.

## 4. Final Remarks

In epithelial tumours, the upregulation of the core EMT TFs (ZEB family, SNAI1/2 and TWIST) and the intermediate filament vimentin are related to the development of different types of invasive adhesions such as degradative focal contacts, invadopodia and microtentacles. In haematological tumours, it remains largely unknown whether the expression of EMT TFs and the development of cytoskeletal and adhesive structures are coordinated similarly to epithelial tumours; or whether the expression changes are constrained in distinctive combinations according to the microenvironment and/or the type of haematological cancer. Notably, the formation of particular invasive adhesions associated with EMT remain largely unstudied in haematological cancers. Given the substantial evidence indicating that the development of EMT signatures correlates with poor prognosis in leukaemia, lymphoma and multiple myeloma, it will be critical in the coming years to determine the adhesions and cytoskeletal organisation associated with this phenotype. Several drugs that may interfere with the signalosomes at adhesion sites and/or cytoskeletal -remodelling are already available and could be used as migrastatics: compounds interfering with the migration of invasive cancers [141,142,143] to improve therapeutic interventions and avoid metastasis and relapse. 

Determining the composition and regulation of EMT-associated adhesions in haematological malignancies will improve our understanding of the development not only of invasive migration but also the mechanisms leading to drug resistance and immunosuppression. It is anticipated that this research will result in the identification of new therapeutic targets that alone or in combination with cytotoxic therapies will allow for the treatment of patients refractory to current therapies. 

## Figures and Tables

**Figure 1 cells-11-00649-f001:**
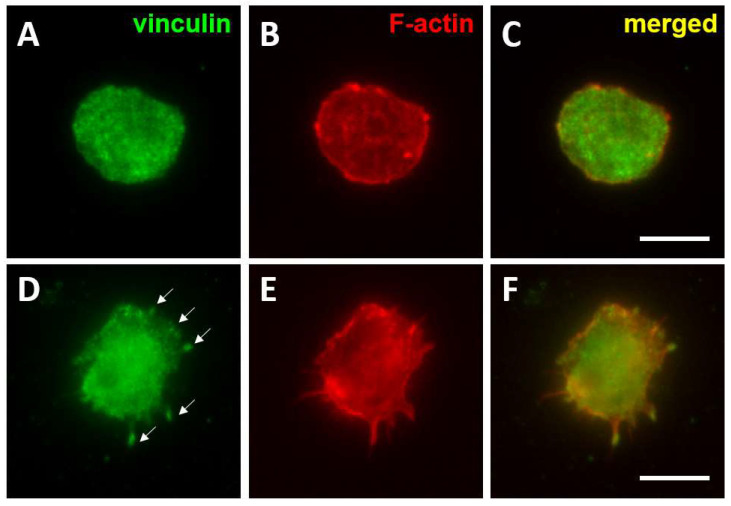
Formation of mature focal contacts in AML cells activated with TPA. THP-1 cells were seeded on fibronectin coated coverslips (10 μg/mL) n and left untreated (**A**–**C**) or were treated with TPA (30 nM for 48 h). Cells were fixed with 4% paraformaldehyde and permeabilised with 0.05% Triton X-100 and immuno-stained to detect the distribution of vinculin (**A**,**D**) and F-actin (**B**,**E**). Merged images are shown in (**C**,**F**). Vinculin was quite homogenously distributed on the surface of untreated THP-1 cells, whereas TPA activation induced the clustering of vinculin into focal contacts (arrows in **D**). Bar 10 μm.

**Figure 2 cells-11-00649-f002:**
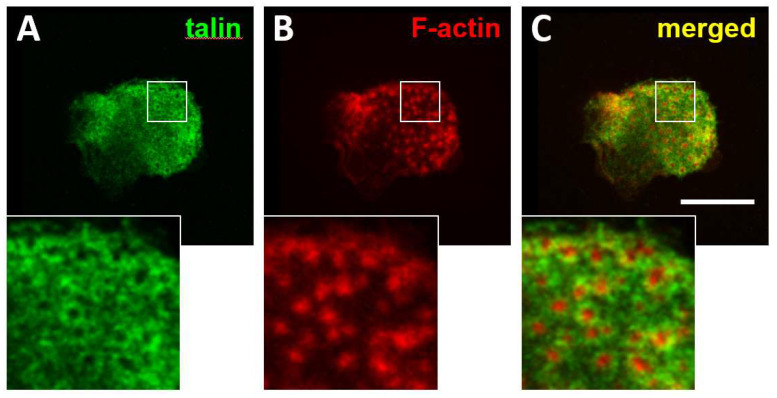
Formation of podosomes in multiple myeloma cells. CD138 positive cells were isolated from bone marrow aspirates from multiple myeloma patients. Isolated cells were seeded on fibronectin-coated coverslips and incubated for 24 h. Cultures were fixed with 4% paraformaldehyde and permeabilised with 0.05% Triton X-100 and immuno-stained to detect the distribution of talin (**A**) and F-actin (**B**). Merged images are shown in (**C**). Magnification of the boxed areas with talin, vinculin and merged staining are shown at the bottom. Bar 10 μm.

**Figure 3 cells-11-00649-f003:**
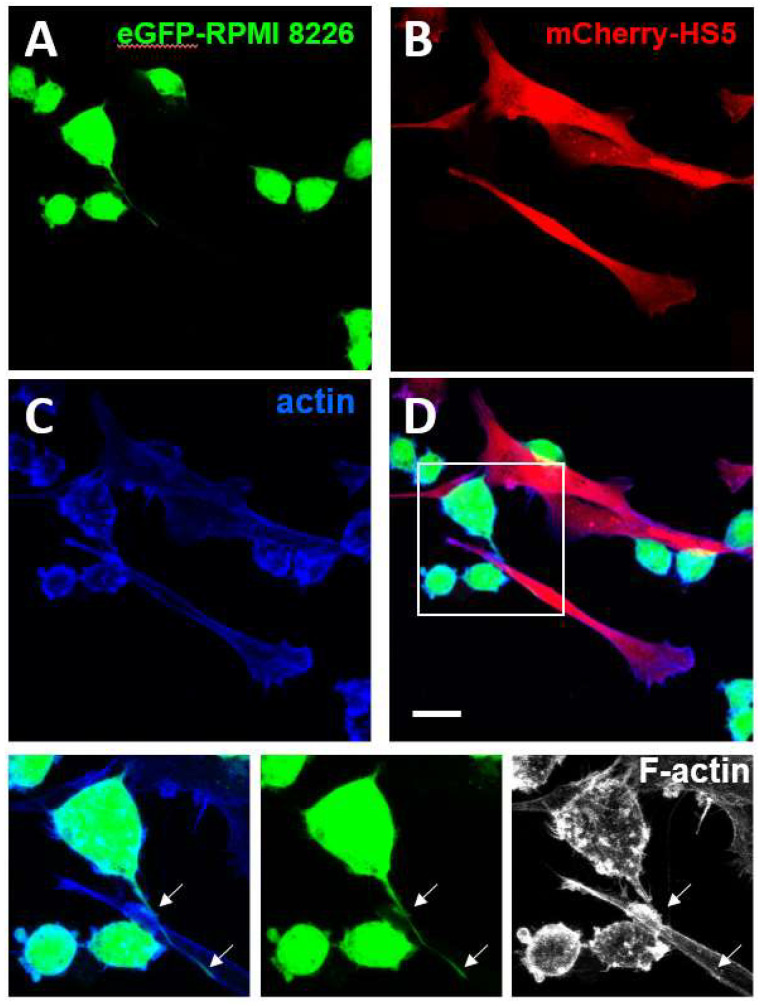
Formation of podia in multiple myeloma cells. The enhanced-Green Fluorescent Protein (eGFP)-expressing multiple myeloma cell line RPMI-8226 (eGFP_RPMI 28226) (**A**) and the monomeric Cherry (m-Cherry)-expressing bone marrow stromal cell line HS5 (mCherry-HS5) (**B**) were generated as previously described [139]. eGFP-RPMI 8226 were co-cultured for three days with mCherry-HS5 cells, fixed with 4% paraformaldehyde and permeabilised with 0.05% Triton X-100 and stained with Alexa 647-Phalloidin to detect the distribution of F-actin (**C**). Merged images are shown in (**D**). Magnification of the boxed areas with eGFP and F-actin staining are shown at the bottom. eGFP-RPMI 8226 cells formed podia (white arrows) elongating along the surface of mCherry-HS5 cells. Bar 10 μm.

**Table 1 cells-11-00649-t001:** Upregulation of canonical EMT markers in haematological malignancies correlates with poor prognosis and/or the enhanced migration of cancer cells.

Mechanism Regulated	EMT TF	Haematological Malignancy Type	References
*Resistance to therapy and poor prognosis*	ZEB1	B-Cell Lymphoma	[45]
		Mantle cell lymphoma	[46]
	TWIST1	CML	[50]
	SNAIL, SLUG	Multiple myeloma	[54]
	ZEB1, HGF	MLL-AF9 AML	[9]
*Enhanced capacity for cell migration*	TWIST1	Paediatric anaplastic	[49]
		large cell lymphoma	/
	ZEB1	MLL-AF9 AML	[9]
	TWIST1, SNAIL, SLUG	Multiple myeloma	[23,51,54]

## Data Availability

Not applicable.

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
