# Peer review of "Epithelial Mesenchymal Transition (EMT) and Associated Invasive Adhesions in Solid and Haematological Tumours"

_cells, 2022, doi:10.3390/cells11040649_

Round 1

Reviewer 1 Report

Overall recommendation:

 Accept

Final comments:

The authors reviewed pathological functions of epithelial–mesenchymal transition (EMT) in solid and haematological tumors. They have provided a comprehensive overview of recently reported clinical and basic research. EMT is observed during both physiological and pathological process. Detailed understanding of molecular mechanisms involved in progression to malignant EMT is fundamental importance in guiding development of effective prevention and treatment.

Author Response

We thank the reviewer for the positive comments about our review.

Reviewer 2 Report

This is a very well-written review by Greaves and Calle that is easy to read and has been very well researched. The review is an excellent overview of EMT focusing on its potential role in hematological malignancies. The authors describe the roles of cell adhesion and attachment in the cell cycle progression and potential implications in EMT. Overall, it is a good and informative review of the literature. Minor points only:

1. Typo - all greek symbols are missing.

2. Although the IFs illustrate nicely how hematological malignant cells adhere to the surface, a table summarizing the principal actors described in section 2 might be relevant for readers. In my opinion, this paragraph is very important and a table showing ZEB1, TWIST1, among other factors in leukemia or MM cells might be useful. 

Author Response

We thank the reviewer for the comments:

1. Typo - all greek symbols are missing.

We apologise for having missed that the greek symbols did not copy from our word document to the mdpi template. This problem has been amended in the new version.

2. Although the IFs illustrate nicely how hematological malignant cells adhere to the surface, a table summarizing the principal actors described in section 2 might be relevant for readers. In my opinion, this paragraph is very important and a table showing ZEB1, TWIST1, among other factors in leukemia or MM cells might be useful. 

Thank you very much for this suggestion. We have now included a table  summarising the EMT markers shown to be involved in the poor prognosis or the enhanced invasive migratory capacity of haematological malignancies. We hope this satisfies the request from the reviewer. We agree this table facilitates the reading of the section by structuring and summarising its content. The added text and table are written in red in the re-submitted version.